# The Effect of Ultra-Late Cranioplasty in a Patient with Long-Term Disorders of Consciousness

**DOI:** 10.3390/brainsci14101038

**Published:** 2024-10-19

**Authors:** Marianna Contrada, Federica Scarfone, Maria Girolama Raso, Lucia Francesca Lucca, Antonio Cerasa, Maria Elena Pugliese

**Affiliations:** 1Sant’Anna Institute, Via Siris 11, 88900 Crotone, Italy; federica.scarfone10@gmail.com (F.S.); m.raso@istitutosantanna.it (M.G.R.); l.lucca@istitutosantanna.it (L.F.L.); me.pugliese@isakr.it (M.E.P.); 2Institute of BioImaging and Complex Biological Systems (IBSBC-CNR), Via T. Campanella, 88100 Catanzaro, Italy

**Keywords:** ultra-late cranioplasty (CP), functional recovery in long term, traumatic brain injury (TBI), vegetative state/unresponsive wakefulness syndrome (VS/UWS), minimally conscious state (MCS), disorder of consciousness (DoC)

## Abstract

Background/Objectives: Cranioplasty (CP) is the main surgical procedure aiming to repair a morphological defect in the skull. It has been shown that early CP is useful for patients with traumatic brain injury (TBI) to achieve functional recovery, whereas few studies have investigated the clinical effects of ultra-late CP on TBI outcomes. Methods: Here, we describe the clinical course over 2 years of a TBI patient who underwent CP 19 months after fronto-parietal decompressive craniectomy (DC) of a limited size. Results: We found that after ultra-late CP, a meaningful functional recovery (cognitive and motor), with emergence from a minimally conscious state and recovery of functional communication, was revealed. Conclusions: Our preliminary findings contribute to the actual debate on the timing of CP for this neurosurgical procedure’s therapeutic success, as early CP has already been shown.

## 1. Introduction

Decompressive craniectomy (DC) is an emergency life-saving surgical procedure used to alleviate increased intracranial pressure (ICP) in patients with severe brain injury. It involves the removal of a portion of the skull to allow the swollen brain to expand without being compressed [1]. This procedure is typically employed in cases of traumatic brain injury (TBI), stroke, or other forms of acquired brain injury where medical management of ICP fails [2]. Cranioplasty (CP), instead, is a follow-up reconstructive procedure performed to repair the skull defect created by DC [3]. This is usually performed weeks to months after DC once brain swelling has subsided. CP involves replacing the removed bone flap or using synthetic materials to cover the skull opening. Its primary goals are to protect the brain, restore the normal appearance of the skull, and potentially improve neurological function [4].

Through the restoration of normal vascular circulation and cerebrospinal fluid, CP is successful in lowering mortality and improving clinical outcomes [2]. However, despite being a routine neurosurgical treatment, CP has a high likelihood of complications [5]. Up to 34% of CP procedures have been associated with occurrences of infection, hygromas, hydrocephalus, seizures, reoperations, cerebral bleeding, bone resorption, flap depression, and wound dehiscence [6]. The timing of CP is another point of contention that may have limited the neurosurgical procedure’s therapeutic success. Indeed, nowadays, there is no consensus about the optimal time intervals between DC and CP interventions. Routinely, late scheduling of CP (3 months after DC) is conventionally preferred, but many surgeons have questioned the benefits of cranial repair at an early stage (1–3 months after DC) due to its impact on outcomes and complications (i.e., postoperative infections) [7].

At the most recent international conference on progress in neuro-traumatology [8], appropriate experts reached a consensus and proposed that the course of CP can be divided into four phases: ultra-early, <6 weeks; early, from 6 weeks to 3 months; intermediate, 3–6 months; and late, >6 months. This has led many clinicians to draw attention to the disparities in outcomes between ultra-early and early interventions, although consistent findings have never reported that could suggest whether to prescribe earlier or later CP in order to increase the likelihood of clinical recovery [4,5]. For instance, Aloraidi et al. [5] reported no differences in the clinical outcome between early and late CP, whereas for Xu et al. [4], early CP could only shorten the amount of time needed for surgery, but it could not lessen patient complications.

Therefore, to delineate the consequences of CP in late TBI after brain contusion better, we evaluated the outcome of CP when it was performed 19 months after FP DC.

## 2. Case Report History

The clinical history and treatment are described in Figure 1. On 1st January 2022, a 26-year-old man fell from a height of four floors due to an intentional attempt. His past medical history was unremarkable except for him previously having been referred for cocaine abuse. He was immediately brought to the emergency facility in an unresponsive state of consciousness (Glasgow Coma Scale (GCS) = 4), requiring intubation and hemodynamic support. He underwent an urgent right fronto-parietal (FP) DC because of intracranial hypertension. The patient was later transferred to the Intensive Care Unit (ICU) of the City Hospital with a diagnosis of vegetative state/unresponsive wakefulness syndrome (VS/UWS; Rancho Los Amigos Level of Cognitive Functioning (RLA-LCF) = 2) [9]. His legal representative gave written informed consent.

On 21 February 2022, the patient was admitted to the Intensive Rehabilitation Unit (IRU). At admission, the patient was awake with no signs of awareness (VS/UWS). He breathed through a tracheostomy. A percutaneous gastrostomy was made to ensure proper caloric intake and internal hydration. His craniectomy scar showed no signs of dehiscence, with optimal healing. However, brain bulging through the craniectomy window was observed.

The legal representative gave written informed consent. This study was approved by the Central Area Calabria Region in Catanzaro (Protocol No. 343, 21 October 2021).

A Computed Tomography (CT) scan on admission revealed evidence of a right fronto-parietal hygroma, a pathological enlargement of the ventricular system, particularly to the right at the occipital horn, and protrusion of the brain parenchyma at the craniectomy site, a hypodense area that affected the right thalamus.

During his neurorehabilitation stay, the patient underwent daily passive mobilization, respiratory rehabilitation in the prone position with cupping, and unimodal sensitive stimulation. His neurologic state was monitored weekly. He was gradually weaned from the tracheostomy.

The patient demonstrated a shift from VS/UWS to a minimally conscious state (MCS; RLA-LCF: 3) during the six months of his IRU stay. In this period, the patient’s caregivers disagreed with the CP intervention due to fear of complications from the surgery. For this reason, in addition to urinary recurrent infections, the patient was not treated with reconstructive CP.

On 9 August 2022, the patient was moved to a Long-Term Care Disorder of Consciousness (DoC) Specialized Unit (LTC) in agreement with his caregivers’ wishes. During the LTC stay, the patient underwent the following rehabilitative treatments: multiprofessional nursing, including activities like monitoring his vital signs, giving him medications, managing and treating the gastrostomy to maintain proper hydration and nutritional intake, etc.; respiratory physiotherapy and mobilization; and speech therapy and observation to maintain and enhance buccal–lingual–facial muscle tonicity and recover the automatism of swallowing. Additionally, depending on his level of cognitive functioning, stimulation with a laryngeal mask, global stimulation, gustatory stimulation, and specific and non-specific neurosensory stimulation of the oral cavity and the peri-buccal area were given. The multisensory stimulation protocol, including visual, auditory, tactile, and emotional stimulation, was created ad hoc with the caregivers’ assistance. In accordance with Wood’s protocol [10], the “sensory regulation” intervention was carried out twice a week in a quiet and controlled setting, where few and simple stimuli, appropriate for the patient’s restricted cognitive abilities, were employed. Furthermore, conditioning techniques were also employed as part of the behavioral interventions [11] in order to improve the frequency, intensity, and length of the desired behavior and eliminate non-functional behaviors. The multiprofessional team who was responsible for patient care, including the physician, physiotherapist and psychologist, persuaded the family of the neurosurgery reintervention. Thus, one year after admission to the LTC, on 17th August 2023, the patient received a custom-made CP implant.

### 2.1. Cranioplasty Intervention

During his LTC stay, the patient was monitored with repeated CT scans. The last brain CT scan carried out before reconstructive CP (Figure 2A) showed further right occipital horn enlargement with persistence of protrusion of the brain parenchyma at the craniectomy site. One month after the CP, a CT scan was repeated, revealing patches of blood-like hyperdensity in the cortico-subcortical location corresponding to the posterior horn of the right lateral ventricle at the surgical site, with persistent significant right ventricle dilatation (Figure 2B).

The patient thus received a custom-made CP implant made of polyether ether ketone (PEEK), MEDPRIN (https://www.medprin.com/, accessed on 11 January 2024) (See Figure 3).

### 2.2. Clinical Assessment

Four distinct time points were used for clinical and behavioral evaluations of his level of consciousness: at IRU admission and during his LTC stay, before CP, following CP, and at 6-month follow-up after the CP. The clinical battery included (a) the Ranchos Los Amigos Levels of Cognitive Functioning (RLA-LCF); (b) the Glasgow Outcome Scale, Extended (GOSE-E); (c) the Disability Rating Scale (DRS); (d) the Barthel Index (BI); (e) the Early Rehabilitation Barthel Index (ERBI); (f) the Glasgow Coma Scale (GCS); (g) the Brief Post-Coma Scale (BPCS); (h) the Coma Recovery Scale, Revised (CRS-r); (i) the Wessex Head Injury Matrix (WHIM); and (j) the Nociception Coma Scale (NCS).

## 3. Results

Table 1 reports the clinical assessments before and after the CP during the clinical course.

Overall, the patient’s clinical condition improved after the CP, as demonstrated by the RLA-LCF, CRS-r, NCS, GCS, WHIM, and BPCS measurements. This result is in line with the ability of the aforementioned scales to investigate consciousness and cognitive functioning specifically. He was able to interact with the outside world, albeit more or less erratically and with fluctuations, and exhibited behavioral signs of awareness and response. Consequently, Exit-MCS (E-MCS) was the diagnosis made following the CP, according to Bruno et al. [12].

We were able to distinguish between VS/UVRS and an MCS in the patient with reduced consciousness thanks to the BPCS evaluation. Given the patient’s poor eye tracking and visual attention before the CP, this value doubled (from 1.5 to 3) after the CP. After the neurosurgical procedure, his eye tracking and visual fixation were completely recovered; however, spontaneous movements of his upper and/or lower limbs remained unaltered.

According to the results of the CRS-r assessment (see Table 2), the patient was able to carry out reproducible movements to give commands; recognize objects; react voluntarily and automatically; vocalize and move his mouth; communicate purposefully; and show signs of awareness and attention.

Specifically, at the beginning, the patient was in a status defined as MCS minus (MCS−), characterized by non-reflex responses such as the localization of nociceptive stimuli, eye tracking, and emotional reactions in response to salient stimuli. After 18 months (July 2023), he evolved into a status defined as MCS plus (MCS+), showing intelligible verbalization, giving yes/no answers, and following simple orders. Two years after the event (February 2024), the patient was defined as having emerged from an MCS (E-MCS), recovering his functional communication and functional use of objects.

The NCS evaluation showed a rise in the score from 5 to 8. In particular, his motor response changed from abnormal posturing (slow, stereotyped flexion or extension of the upper and/or lower extremities) to flexion withdrawal (isolated flexion withdrawal of at least one limb; the limb must move away from the point of stimulation) in response to the proprioception of pain, while his facial expression remained unchanged. His auditory response shifted from groaning (which is defined as groans that are not spontaneous) to vocalization/oral movement (that is, from at least one instance of non-reflexive oral movement or vocalization in response to stimulation), and his visual response shifted from eye movements (which are anarchical eye movements in response to noxious stimulation) to fixation (which is defined as shifting from one’s initial fixation point and fixating on the examiner for longer than two seconds).

The WHIM examination also confirmed an increase in this score from 49/30 to 57/41. The patient proved that he could disregard distractions (such as averting his attention from a conversation to attend to someone else), mimic a gesture when asked to do so, and indicate understanding by nodding his head or making other movements in accurately responding to nine out of ten questions; find a specific playing card from a group of four and select it correctly nine times out of ten; report the time of day using binary answers of yes and no; and use his eyes to point between two images, real objects, or cards and indicate the ones with the right answers nine times out of ten.

However, no evident clinical improvements were detected by the Barthel index, the DRS, the ERBI, or the GOS-E. This observation was also predictable because these are predominantly disability scales that score patients’ motor functional capability. The patient was able to interact with the outside world, albeit erratically and with fluctuations, and exhibited behavioral signs of awareness and response.

## 4. Discussion

In this study, we observed that even after more than 19 months after FP DC of a limited size, CP could induce a relevant functional recovery, with emergence from an MCS. In particular, the clinical and cognitive improvement was characterized by a meaningful functional recovery (cognitive and motor) with emergence from an MCS and a more focused and defined response (avoidance of pain, blinking upon intense light stimulation, head turning toward sound, and fixing and following others with one’s gaze); reactions to uncomfortable situations (exhibit oppositional attitudes and more pertinent facial expressions); and the ability to comply with both simple and complex commands. The clinical variation we noted was mainly cognitive variation, with severe residual motor disability. In fact, although he emerged from an MCS, the patient continued to be dependent in his daily living activities.

Due to their complexity and multifactorial nature, the precise processes underlying this effect—which include changes in cerebral hemodynamics, metabolism, cerebrospinal fluid (CSF) dynamics, and neuronal function—remain unclear. First, CP enhances brain compliance (the brain’s capacity to adapt to changes in volume) and normalizes intracranial pressure by replacing the protective covering of the skull. Numerous studies have shown that this restoration may prevent symptoms such as headaches, motor deficits, and cognitive impairments and lessen the negative effect of pressure, regardless of when the intervention is initiated [13,14,15]. Additionally, cerebral perfusion pressure (CPP), which regulates blood flow to the brain, is helped to normalize by CP. CP raises the cerebral blood flow (CBF) and CPP, which helps injured brain tissue receive more oxygen and glucose. This improved perfusion promotes recovery by optimizing the metabolic environment for neural repair and recovery. Lastly, CP aids in the restoration of regular CSF reabsorption and circulation. This promotes better brain biomechanics, which enhances the functionality of neural circuits and lowers the risk of hydrocephalus [15,16].

Our study has confirmed the consensus on a lack of a definitive association between the time intervals of CP interventions and their clinical outcomes [8]. Research has shown that CP can be performed weeks or months following a craniectomy. In general, CP is carried out 3–6 months after DC, or later if the surgical site becomes infected [12]. We confirmed that an ultra-late CP intervention could also be performed in a patient with a severe TBI, who showed relevant clinical improvement. Similarities in the neurological outcomes between TBI patients who underwent early and late CP were reported by Aloraidi [5], who described statistically insignificant differences in the rates of overall postoperative complications between early and late CP. Iaccarino et al. [8] confirmed this finding, highlighting that the odds of infections, reoperations, intracranial hemorrhage, and seizures did not appear to be different between early and late CP. Similarly, Corallo et al. [13] demonstrated no significant difference between early CP patients and ultra-late CP patients. This latter finding contrasts with other studies that have suggested prescribing earlier CP in order to increase the likelihood of clinical recovery [3,15]. Nevertheless, Archavlis et al. [16] found that deep wound infections and osteomyelitis seem to be more common in people with early CP and several concomitant illnesses. In summary, the long-term prospective design of our study—in which the patient was followed for more than two years in order to assess every clinical change before and after CP better—should be emphasized in comparison to earlier research. However, results provided by studying a single patient should be taken with caution, and more research with a bigger sample size is needed to really assess the association between varying the timing of CP therapies (from ultra-early to ultra-late) and their clinical outcomes better.

## 5. Conclusions

According to our preliminary findings, ultra-late CP may be used to enhance the clinical outcomes in long-term TBI cases. Since there is a limited amount of literature on this field of study, our work could inspire multicentric research re-examining the relationship between the timing of CP and neurological outcomes in light of its preliminary significance.

## Figures and Tables

**Figure 1 brainsci-14-01038-f001:**
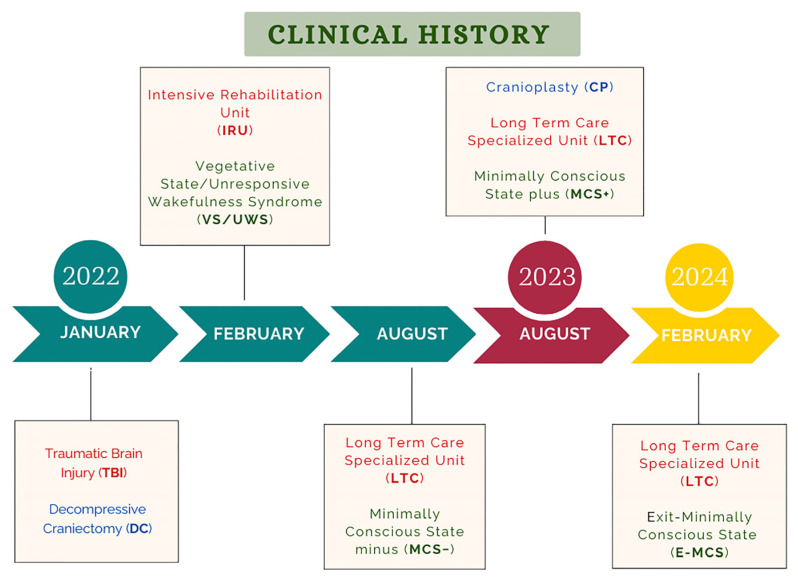
Clinical history and treatment. In January 2022, after brain contusion, this TBI patient underwent FP DC therapy before being admitted to the IRU in a VS/UWS status. Following this initial phase, the patient was transferred to an LTC, where, following CP, his condition improved from MCS− to E-MCS.

**Figure 2 brainsci-14-01038-f002:**
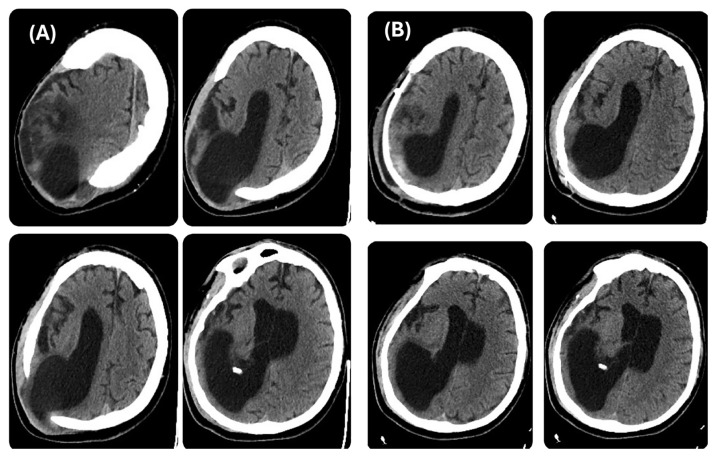
Brain CT scan, axial view. Left side (**A**) before cranioplasty and right side (**B**) after cranioplasty.

**Figure 3 brainsci-14-01038-f003:**
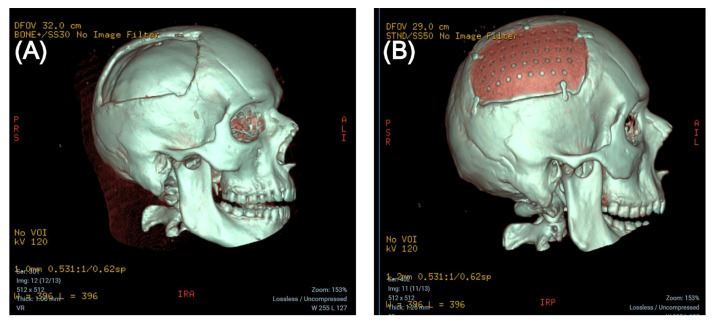
Three-dimensional CT scan before (**A**) and after (**B**) cranioplasty.

**Table 1 brainsci-14-01038-t001:** Timeline of clinical assessments.

February 2022 (IRU Admission)	August 2022 (LTC Admission)	July 2023 (LTC Stay Pre-CP)	September 2023 (LTC Stay Post-CP)	February 2024 (LTC Stay Follow-Up)
RLA-LCF: 2	RLA-LCF: 3	RLA-LCF: 3	RLA-LCF: 3	RLA-LCF: 5/6
GOS-E: 2	GOS-E: 3	GOS-E: 3	GOS-E: 3	GOS-E: 3
DRS: 23	DRS: 21	DRS: 21	DRS: 21	DRS: 21
GCS: 7	GCS: 7	GCS: 7	GCS: 12	GCS: 12
BPCS: 1.5	BPCS: 1.5	BPCS: 1.5	BPCS: 3	BPCS: 3
CRS-r: 6	CRS-r: 7	CRS-r: 13	CRS-r: 15	CRS-r: 21
NCS: 3	NCS: 3	NCS: 5	NCS: 6	NCS: 8
WHIM: N/A	WHIM: 17 (11)	WHIM: 49 (29)	WHIM: 49 (33)	WHIM: 57 (41)
BARTHEL: 0	BARTHEL: 0	BARTHEL: 0	BARTHEL: 0	BARTHEL: 0
ERBI: −175	ERBI: −125	ERBI: −125	ERBI: −125	ERBI: −125

IRU (Intensive Rehabilitation Unit); LTC (Long-Term Care Specialized Unit); RLA-LCF (Ranchos Los Amigos Levels of Cognitive Functioning); GOS-E (Glasgow Outcome Scale, Extended); DRS (Disability Rating Scale); GCS (Glasgow Coma Scale); BPCS (Brief Post-Coma Scale); CRS-r (Coma Recovery Scale, Revised); NCS (Nociception Coma Scale); WHIM (Wessex Head Injury Matrix); BARTHEL (the Barthel Index); ERBI (Early Rehabilitation Barthel Index).

**Table 2 brainsci-14-01038-t002:** Timeline of CRS-r sub-item assessments.

	February 2022 (IRU Admission)	August 2022 (LTC Admission)	July 2023 (LTC Stay Pre-CP)	September 2023 (LTC Stay Post-CP)	February 2024 (LTC Stay Follow-Up)
	CRS-r: 6	CRS-r: 7	CRS-r: 13	CRS-r: 15	CRS-r: 21
1. Auditory	1	1	3	3	4
2. Visual	2	3	3	3	5
3. Motor	0	0	2	3	6
4. Oromotor–verbal	1	1	2	2	2
5. Communication	0	0	0	1	1
6. Arousal	2	2	3	3	3

Diagnosis	VS/UWS	MCS−	MCS+	MCS+	E-MCS

CRS-r (Coma Recovery Scale, Revised).

## Data Availability

The raw data supporting the conclusions of this article will be made available by the authors on request due to due to privacy and ethical restrictions.

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
