# Peer review of "The Effect of Ultra-Late Cranioplasty in a Patient with Long-Term Disorders of Consciousness"

_brainsci, 2024, doi:10.3390/brainsci14101038_

Round 1
Reviewer 1 Report
Comments and Suggestions for Authors
The authors are submitting the manuscript titled The Effect of Ultra-Late Cranioplasty in a Patient with Long-Term Disorders of Consciousness. It is in the format of a case report. The positive impact of ultra-late cranioplasty on the clinical outcome of a TBI patient is documented through a single clinical observation. This is a relatively rare and well-documented case. Conclusive literary data on the positive effect of ultra-late cranioplasty are lacking.
I have the following comments regarding the manuscript:
- The manuscript lacks a specification of the type of TBI (brain contusion, acute subdural hematoma, diffuse axonal injury, etc.). This is essential and needs to be added.
- The decompressive craniectomy documented by CT has several shortcomings. The craniectomy is not large enough to achieve its decompressive goal and is located in an atypical position. The temporal region is not involved in the decompression at all. Therefore, I would not call this an FTP decompressive craniectomy, but rather an FP decompressive craniectomy of limited size.
- Several scoring scales are used in the manuscript to document the improvement in clinical status. Since this was a case of TBI, it would be appropriate to also include the Ranchos Los Amigos Cognitive Scale.
- The discussion lacks hypotheses and an explanation for the surprising clinical improvement after cranioplasty, which was performed after such a long time. The patient showed improvement despite being in a clinically unfavorable prognostic condition. This is necessary to add to the discussion.
Author Response
The authors are submitting the manuscript titled The Effect of Ultra-Late Cranioplasty in a Patient with Long-Term Disorders of Consciousness. It is in the format of a case report. The positive impact of ultra-late cranioplasty on the clinical outcome of a TBI patient is documented through a single clinical observation. This is a relatively rare and well-documented case. Conclusive literary data on the positive effect of ultra-late cranioplasty are lacking.
I have the following comments regarding the manuscript:
- The manuscript lacks a specification of the type of TBI (brain contusion, acute subdural hematoma, diffuse axonal injury, etc.). This is essential and needs to be added.
REPLY: This is a brain contusion TBI. This information has been included in Line n° 58.
2. The decompressive craniectomy documented by CT has several shortcomings. The craniectomy is not large enough to achieve its decompressive goal and is located in an atypical position. The temporal region is not involved in the decompression at all. Therefore, I would not call this an FTP decompressive craniectomy, but rather an FP decompressive craniectomy of limited size.
REPLY: Following reviewer’s suggestion we modified this term.
3. Several scoring scales are used in the manuscript to document the improvement in clinical status. Since this was a case of TBI, it would be appropriate to also include the Ranchos Los Amigos Cognitive Scale.
REPLY: This additional information has been reported in the Line n. 67, 68, as well as, in the table 1.
4. The discussion lacks hypotheses and an explanation for the surprising clinical improvement after cranioplasty, which was performed after such a long time. The patient showed improvement despite being in a clinically unfavorable prognostic condition. This is necessary to add to the discussion.
REPLY: we would like to thank this reviewer for highlighting this important point. In the discussion a new section has been added in order to better explain for this surprising clinical improvement. Line 212-224
Reviewer 2 Report
Comments and Suggestions for Authors
Dear Authors,
Thank you for the opportunity to review your manuscript on the impact of delayed cranioplasty in a traumatic brain injury patient. I appreciate the important insights you provide, particularly regarding the potential for meaningful recovery even many months post-decompression. Your work contributes significantly to the ongoing discussions about optimal timing for surgical interventions in neurocritical care.
Abstract
- Clarity and Conciseness: The abstract could benefit from a more concise structure. Consider reducing the number of sentences and combining related ideas. For instance, the mention of "the timing of CP affect outcome has never been solved" could be rephrased to "the effect of CP timing on outcomes remains unresolved."
- Terminology: Use consistent terminology for clarity. For example, you switch between "delayed CP" and "ultra-late CP." Choose one term and use it consistently.
- Specificity: When stating "meaningful functional recovery," briefly define what that entails—specific improvements in cognitive and motor functions would provide clearer insights.
Introduction
- Transitions: The transition between sections could be smoother. For example, after discussing DC, introduce CP more fluidly, explaining how it relates to the prior procedure.
- Citations: Ensure that all citations are in the appropriate format and appear consistently throughout the manuscript.
- Objective Statement: Clearly state the study's objective at the end of the introduction. Something like, "This case report aims to evaluate the outcomes of cranioplasty performed 19 months post-decompressive craniectomy in a TBI patient."
Case Report
- Details on Procedures: Provide more detail about the specific rehabilitative interventions the patient underwent, especially the rationale behind them.
- Visuals: Ensure figures are referenced in a way that adds value to the text. For example, mention what specific information the reader should glean from Figure 1.
Results
- Clarity of Data Presentation: The results section could benefit from more explicit connections between data points and their clinical significance. Instead of just stating the scores, briefly explain what the changes indicate in terms of patient progress.
- Use of Tables: While Table 1 is helpful, consider adding a brief summary paragraph that encapsulates the key findings from the table, emphasizing improvements and their implications.
Discussion
- Contextualization: Discuss your findings in relation to existing literature more explicitly. For example, how do your results compare with those of Aloraidi or Iaccarino?
- Implications: Clearly articulate the clinical implications of your findings. What do they mean for future practice regarding the timing of CP?
- Limitations: Acknowledge any limitations of your case report, such as the single-patient design or potential biases in interpretation.
Conclusions
- Strength of Conclusion: Reinforce the conclusions by tying them back to the broader implications for clinical practice.
- Future Directions: Consider suggesting areas for future research, such as studies involving larger cohorts or controlled trials examining the timing of CP in TBI patients.
General Suggestions
- Proofreading: Check for typographical errors and grammatical inconsistencies. For example, “has been shown that CP can be useful” should be “has been shown that CP is useful.”
- Style Consistency: Ensure that terminology and style are consistent throughout the manuscript. For instance, use either “TBI” or “Traumatic Brain Injury” throughout, not both interchangeably.
- Keywords: Consider including additional relevant keywords that might enhance discoverability.
Overall, your manuscript provides valuable insights into the benefits of delayed cranioplasty and its potential to improve outcomes for TBI patients. With these revisions, I believe it can make an even greater impact in the field. I look forward to seeing the final version!
Author Response
Dear Authors,
Thank you for the opportunity to review your manuscript on the impact of delayed cranioplasty in a traumatic brain injury patient. I appreciate the important insights you provide, particularly regarding the potential for meaningful recovery even many months post-decompression. Your work contributes significantly to the ongoing discussions about optimal timing for surgical interventions in neurocritical care.
Abstract
1. Clarity and Conciseness: The abstract could benefit from a more concise structure. Consider reducing the number of sentences and combining related ideas. For instance, the mention of "the timing of CP affect outcome has never been solved" could be rephrased to "the effect of CP timing on outcomes remains unresolved."
Reply: the Abstract has been reformulated following the reviewer’s suggestion.
2. Terminology: Use consistent terminology for clarity. For example, you switch between "delayed CP" and "ultra-late CP." Choose one term and use it consistently.
Reply: Done
3. Specificity: When stating "meaningful functional recovery," briefly define what that entails—specific improvements in cognitive and motor functions would provide clearer insights.
Reply: In order to better describe the functional recovery of our patient we now included a new table 2, where the timeline of CRS-r assessment was described.
Introduction
1. Transitions: The transition between sections could be smoother. For example, after discussing DC, introduce CP more fluidly, explaining how it relates to the prior procedure.
Reply: The introduction has been reformulated following the reviewer’s suggestion
2. Citations: Ensure that all citations are in the appropriate format and appear consistently throughout the manuscript.
Reply: DONE
3. Objective Statement: Clearly state the study's objective at the end of the introduction. Something like, "This case report aims to evaluate the outcomes of cranioplasty performed 19 months post-decompressive craniectomy in a TBI patient."
Done.
Case Report
1. Details on Procedures: Provide more detail about the specific rehabilitative interventions the patient underwent, especially the rationale behind them.
Reply: This section has been improved. See Line n.108-113.
2. Visuals: Ensure figures are referenced in a way that adds value to the text. For example, mention what specific information the reader should glean from Figure 1.
Reply: done. See figure 1.
Results
- Clarity of Data Presentation: The results section could benefit from more explicit connections between data points and their clinical significance. Instead of just stating the scores, briefly explain what the changes indicate in terms of patient progress.
- Use of Tables: While Table 1 is helpful, consider adding a brief summary paragraph that encapsulates the key findings from the table, emphasizing improvements and their implications.
Reply: As concerns the results section, we explicitly reported the main scores of clinical data and, next, we discuss the clinical relevance of these data.
Discussion
1. Contextualization: Discuss your findings in relation to existing literature more explicitly. For example, how do your results compare with those of Aloraidi or Iaccarino?
Reply: A new statement has been included. See 241-246
2. Implications: Clearly articulate the clinical implications of your findings. What do they mean for future practice regarding the timing of CP?
Reply: a new section has been added in the conclusions. See line n°254-260
3. Limitations: Acknowledge any limitations of your case report, such as the single-patient design or potential biases in interpretation.
Reply: Done see line 242-244
Conclusions
1. Strength of Conclusion: Reinforce the conclusions by tying them back to the broader implications for clinical practice.
Reply: the clinical relevance of our study has been already stressed. See line n°254-260
2. Future Directions: Consider suggesting areas for future research, such as studies involving larger cohorts or controlled trials examining the timing of CP in TBI patients.
Reply: a new section has been added in the conclusions. See line n°254-260
General Suggestions
- Proofreading: Check for typographical errors and grammatical inconsistencies. For example, “has been shown that CP can be useful” should be “has been shown that CP is useful.”
Reply: Done.
- Style Consistency: Ensure that terminology and style are consistent throughout the manuscript. For instance, use either “TBI” or “Traumatic Brain Injury” throughout, not both interchangeably.
Reply: Done.
- Keywords: Consider including additional relevant keywords that might enhance discoverability.
Reply: Done.
Overall, your manuscript provides valuable insights into the benefits of delayed cranioplasty and its potential to improve outcomes for TBI patients. With these revisions, I believe it can make an even greater impact in the field. I look forward to seeing the final version!
Reply: We would like to express our appreciation for the reviewer’ comments. We feel that our manuscript is strongly improved by incorporating its suggestions.
Reviewer 3 Report
Comments and Suggestions for Authors
I would suggest including CRS-R subscale scores along with the level of consciousness (comatose, vegetative, minimally conscious, emerged states) in table 1 since the total scores only do not mean much (i.e. a patient can have a lower total CRS-R score but in MCS vs. another patient who scored higher but remain in VS). CRS-R score 13 prior to CP and CRS-R score 15 after CP does not look significantly different without the subscales. Curious to know if patient's behavioral signs of consciousness improved after CP. Also recommend that the paragraph about CRS-R (151-154) is elaborated more in terms of what the patient could do prior to CP and what they could do after CP.
It was not too clear what "repetitive actions" meant in 152. Did it mean the patient showed reproducible movement to command or reflexive movements?
Author Response
1. I would suggest including CRS-R subscale scores along with the level of consciousness (comatose, vegetative, minimally conscious, emerged states) in table 1 since the total scores only do not mean much (i.e. a patient can have a lower total CRS-R score but in MCS vs. another patient who scored higher but remain in VS). CRS-R score 13 prior to CP and CRS-R score 15 after CP does not look significantly different without the subscales. Curious to know if patient's behavioral signs of consciousness improved after CP. Also recommend that the paragraph about CRS-R (151-154) is elaborated more in terms of what the patient could do prior to CP and what they could do after CP.
Reply: We would like to thank this reviewer for this important suggestion. A new table 2 has been added reporting the CRS-r relevant single scores.
It was not too clear what "repetitive actions" meant in 152. Did it mean the patient showed reproducible movement to command or reflexive movements?
REPLY: Line 162. Done. We have replaced “repetitive actions” with “reproducible movement to command”
Round 2
Reviewer 1 Report
Comments and Suggestions for Authors The authors have adequately addressed my comments, based on which they have revised and supplemented the manuscript. In its current form, I recommend the manuscript for publication.
Author Response
Dear Editor
According to your suggestion the abstract and the conclusions have been reformulated. Thanks a lot for your assistance.